# METADIST: AN INFRASTRUCTURE FOR AUTOMATIC PARALLELISM VIA SHARDCOMBINE ALGORITHM

## ABSTRACT

As models become larger and hardware limitations widen, parallel training techniques have become increasingly important for improving training efficiency. However, the choice and combination of these techniques can greatly impact their effectiveness. Automatic parallelism methods have emerged to select the best combination of strategies from a selection space of parallel strategies. However, these methods rely heavily on manual annotation of operator SPMD sharding rules, which makes them difficult to develop, maintain and benchmark, and lacking in ecological compatibility. In this work, we present MetaDist, an infrastructure for automatic parallelism. We propose two abstract data structures, MetaOp and MetaIR, which enable us to construct the MetaSPMD space. The ShardCombine Algorithm obviates the need for manual annotation, significantly reducing the development and maintenance cost. Moreover, our approach is natively compatible with multiple ecologies, including PyTorch and JAX. To validate our design, we implement two baseline automatic parallelism algorithms based on MetaDist. Our experiments demonstrate that our approach achieves state-of-the-art performance compared with other distributed solutions.

## 1 INTRODUCTION

Deep learning has achieved significant progress in the quest for artificial general intelligence, and its success can largely be attributed to the development of foundational models (Bommasani et al., 2021) that can scale up with increasing amounts of data and parameters. As these foundational models grow in size, they exhibit significant improvements in performance for downstream tasks and undergo emergence of new capabilities. Because of the rapid growth of models, and the huge gap between hardware limitations, a series of parallel training efforts have emerged, including tensor parallelism (Shazeer et al., 2018; Shoeybi et al., 2019; Karakus et al., 2021), pipeline parallelism (Huang et al., 2019; Fan et al., 2021; Li & Hoefler, 2021), etc. The training of large models is often a combination of these parallel methods, and different strategies can lead to huge performance differences (Narayanan et al., 2021). Existing large model training frameworks still require complex developing involving high performance computing experts to have good performance.

Automatic parallelism holds great potential for accelerating deep learning research and production by allowing model developers to quickly explore new designs without worrying about distributed strategies or performance optimizations. There are two main approaches to automatic parallelism: the template-based approach (Chen et al., 2023; Miao et al., 2022) and the compiler approach (Zheng et al., 2022; Zhang et al., 2023). The template-based approach is limited to specific model structures, such as the transformer architecture, and relies on predefined templates for parallelization. On the other hand, the compiler approach offers more flexibility and can be applied to any model structure, but relies heavily on manual annotation of operator SPMD sharding rules, which involves enumerating the feasible parallelism of each operator. These annotations are often specific to a particular intermediate representation (IR), such as Alpa (Zheng et al., 2022) and Rhino's (Zhang et al., 2023) SPMD annotations for XLA (Sabne, 2020) operators. The coupling between the automatic parallelism algorithm and the IR and its ecological framework leads to two main challenges:

**1) Lack of ecological compatibility:** Automatic parallelism algorithms developed for one framework may not be directly applicable to other frameworks, resulting in duplicated work within the

community. This lack of compatibility limits the reusability of parallelization algorithms across different deep learning frameworks, making it harder to take advantage of existing solutions.

**2) Difficulty in development, maintenance, and benchmarking:** Automatic parallelism algorithms that are strongly coupled to specific IRs and frameworks can be challenging to develop, maintain, and benchmark. As IRs and frameworks evolve over time, the automatic parallelism algorithms need to be updated accordingly, leading to additional development and maintenance efforts. Furthermore, benchmarking and comparing the performance of different parallelization algorithms becomes complicated due to the dependence on specific IRs and frameworks.

Our key observation and motivation revolve around the idea that by automatically exploring SPMD rules for arbitrary operators, we can decouple automatic parallel algorithms from specific IR or machine learning (ML) frameworks. This approach allows us to eliminate the need for manually annotating, reducing the cost of developing and maintaining sharding annotations. Additionally, it enables us to use a set of automatically parallel algorithms across different ML framework ecosystems, making benchmarks between these algorithms more fair.

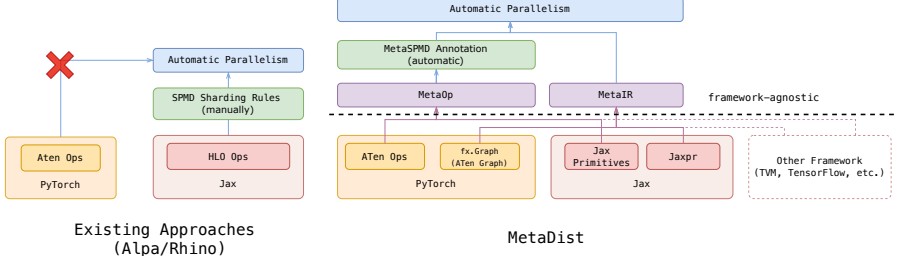

Figure 1: The architecture of existing approaches and MetaDist. MetaDist decouples automatic parallelism from specific IR and frameworks (MetaOp and MetaIR, Section 3.2) and propose an automatic annotation algorithm (ShardCombine Algorithm, Section 3.3) for operator SPMD sharding rules without the need for manual annotations, as often required in existing approaches.

With the aforementioned observation and motivation, we present MetaDist, an infrastructure for automatic parallelism utilizing the ShardCombine Algorithm. To achieve this, we introduce two abstraction structures: MetaOp and MetaIR. MetaOp serves as an abstraction for operators from different frameworks, encompassing kernels from compiler codegen, and supporting *PyTorch ATen* (Paszke et al., 2019), *Jax Primitives* (Frostig et al., 2018), and *TVM Tensor Expression* (Chen et al., 2018). MetaIR unifies a framework-agnostic representation of computational graphs, accommodating *PyTorch ATen graph* (Wu, 2023) and *Jaxprs*. At the abstraction level of MetaOp, we construct the MetaSPMD (Meta Single Program Multiple Data) space and propose the ShardCombine Algorithm, which automates generate MetaSPMD annotation. Finally, we implement two baseline automatic parallelism algorithms based on MetaDist to validate the design and implementation.

## 2 BACKGROUND

### 2.1 DISTRIBUTED ALGORITHM AND SYSTEM

Data parallelism has been the go-to approach for distributed training for a long time. It involves splitting the data across different devices and performing local forward and backward computations separately, followed by gradient synchronization using all-reduce communication. However, as it requires a complete copy of the parameters on each device, it's not suitable for training large models. To address this issue, tensor parallelism and pipeline parallelism have been proposed, with the main difference being the way parameters are partitioned. Tensor parallelism divides the parameters within a layer, while pipeline parallelism divides them between layers. These approaches enable the parallel training of large models by reducing the memory requirements on each device.

Megatron-LM leverages a hybrid approach called PTD Parallelism, which combines pipeline parallelism, tensor parallelism, and data parallelism. To use PTD Parallelism, users need to modify their model to use layers that support tensor parallelism and adjust the ratio of individual parallelisms to

achieve optimal performance. Meanwhile, developers need to implement tensor parallelism layers for and manually manage computation and communication.

Another approach is the ZeRO Redundancy Optimizer (ZeRO) Rajbhandari et al. (2020), which adapts data parallelism by sharding the model parameters, gradients, and optimizer states. From the user's perspective, it is easier to use than other methods, but still requires strategic choices (ZeRO-2 or ZeRO-3) to obtain the best performance. From the developer's perspective, ZeRO's implementation relies heavily on PyTorch's Module feature, which requires manual scheduling and communication management, making it difficult to port to TensorFlow Abadi et al. (2016) or Jax.

Recent automatic parallelism work, such as Alpa Zheng et al. (2022) and Rhino Zhang et al. (2023), abstract parallelism problems as optimization problems to find the least costly solution for communication by using Integer Linear Programming (ILP) or Dynamic Programming (DP). Such a system requires no user-set policies and is easy to use. However, from a development point of view, this class of work requires manual annotations of SPMD rules for each type of operator (typically tens to hundreds) in a particular IR. Such systems are not framework-agnostic, and implementing an identical set of automatic parallelism algorithms for each framework requires significant annotating and development costs, as well as maintenance costs as the framework evolves.

Table 1: Comparison of current distributed solution for large-scale deep learning.

| Distributed Solution | Large Model Support | Easy-to-use (user's view) | Easy-to-dev (develop's view) | Ecological Compatibility |
|---|---|---|---|---|
| PTD Parallelism | 👍 | 👎 | 👎 | 👎 |
| ZeRO | 👍 | 👍 | 👎 | 👎 |
| Alpa/Rhino | 👍 | 👍 | 👎 | 👎 |
| MetaDist | 👍 | 👍 | 👍 | 👍 |

## 2.2 MACHINE LEARNING FRAMEWORK

The current machine learning frameworks can typically be classified into two categories: *eager mode* and *graph mode*, which correspond to imperative and declarative programming, respectively. Most modern frameworks support both modes. For instance, Jax uses eager execution by default for `jax.Array` calculations, but it provides Just In Time (JIT) compilation for better performance. This feature converts Python functions to Jaxpr, and lowers it to XLA. Jax uses HLO (High Level Operations) IR to construct a computational graph for analysis and optimization. The HLO operator is more fine-grained, containing only a few dozen operators.

PyTorch 2.0 introduced a compiler infrastructure that supports graph mode. AOT Autograd traces the forward and backward graphs ahead of time, builds a computational graph based on the PyTorch ATen operator, and provides an API for optimizing and transforming the graph. Moreover, PyTorch's newly introduced DTensor (Distributed Tensor) provides programming primitives and run-time for the SPMD paradigm.

## 3 METADIST

### 3.1 THE OBSERVATION

After observing the current machine learning frameworks and operator-level SPMD parallelism, we can identify two main patterns:

**1) As mentioned in Section 2.2, all machine learning frameworks utilize similar computational graph structures and operators.** Operators is a set of fundamental operations in deep learning, such as convolution and layer normalization. Computational graphs are comprised of tensors and operators. In computational graphs, nodes typically represent operators, and directed edges between nodes depict tensor states and signify the dependencies between computations. For instance, Jax employs Jaxpr as the computational graph and Primitives as operators, while PyTorch uses the FX graph traced by AOT Autograd as the computational graph and ATen operators.

**2) The common pattern of SPMD operation rules involve: specific sharding input, local computation, re-combineable output**, as illustrated in the right portion of Figure 2. For example, to perform the matrix multiplication $Y = XW$, the first dimension of $X$ is sharded, $W$ is replicated, and local matrix multiplication is executed, resulting in local $Y$. Gathering local $Y$ in the first dimension produces the global $Y$.

Building upon the two aforementioned observations, we have developed two framework-agnostic concepts: MetaIR and MetaOp. Computational graphs from various frameworks can be transformed into MetaIR, wherein the operators are converted into MetaOp. The ShardCombine algorithm is then utilized to automatically annotate SPMD rules on MetaOp, facilitating the design and implementation of parallel algorithms based on MetaOp and MetaIR.

## 3.2 METAOP AND METAIR

MetaIR provides an abstract representation of computational graphs from diverse frameworks, comprising of three parts: `input_list`, `op_list`, and `output_list`. The `input_list` and `output_list` consist of several MetaVars, which serve as abstract representations of tensors. MetaVars contain information such as the shape and data type of tensors, as illustrated in Figure 2. The `op_list` comprises of numerous MetaOp, each of which receives a set of MetaVars as input and produces a set of MetaVars as output. Additionally, MetaOp contains a callable primitive operator function from the frameworks, as well as MetaSPMD, which encompasses the operator parallelism space.

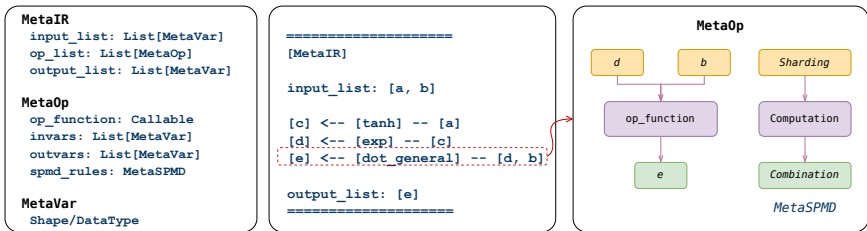

Figure 2: The definition and example for MetaIR and MetaOp.

MetaIR can be considered as a directed acyclic graph (DAG), with MetaOp serving as nodes and MetaVar as edges. The central portion of Figure 2 depicts a simple instance of MetaIR, which has two MetaVars (`a, b`) as input and a single MetaVar (`e`) as output, along with three MetaOps (`tanh, exp, dot_general`). However, since the granularity of operators differs across various frameworks, the MetaIR's general abstraction would differ for an equivalent computation. In the case of PyTorch's ATen operator, which has a coarser granularity, its corresponding MetaIR usually features a smaller number of MetaOps. Conversely, Jax, which has a finer granularity, typically incorporates a greater number of MetaOps in its MetaIR representation. And we will describe how to make MetaOp support TVM Tensor Expression in the Appendix A.1.

## 3.3 METASPMD AND SHARDCOMBINE ALGORITHM

In accordance with our second observation in Section 3.1, we have designed MetaSPMD as the part of MetaOp that expresses the operator-level parallel space. MetaSPMD consists of a ShardSpec and a CombineSpec that detail how to shard the input and combine the local results into global results, respectively. For a particular MetaOp, let us assume that it has $i$ inputs, denoted by $X_1$, $X_2$, ..., $X_i$, where each input $X_i$ has a tensor shape of $[D_{i_1}, D_{i_2}, ..., D_{i_n}]$ (i.e., tensor $X_i$ has $i_n$ dimensions). The ShardSpec takes a list of $i_n$ values, $[C_{i_1}, C_{i_2}, ..., C_{i_n}]$, for each input $X_i$, where each value corresponds to a dimension of the tensor $X_i$ and takes on the values `NoShardDim` ($N$) or `ShardDim(id=j)` ($S_j$). The value `NoShardDim` signifies that the dimension is not shardable, while $S_j$ corresponds to dimensions that can be sharded simultaneously. For each $S_j$, there is a corresponding `CombineFunc` that can re-combine the local results into global results. The CombineSpec is a dictionary whose key is $S_j$, and its value is its corresponding `CombineFunc`. Common `CombineFunc` include `gather`, `reduce`, and so on.

Figure 3 illustrates some examples of the MetaSPMD for common operators. In the case of the matrix multiplication (MatMul) operator, there are two inputs with ShardSpecs $[S_1, S_2]$ and $[S_2, S_3]$, indicating three sharding strategies: $S_1$ (sharding the first dimension of the first input, corresponding to data parallelism), $S_2$ (sharding the first input of the second dimension and the first dimension of the second input, which corresponds to row tensor parallelism in Megatron-LM), and $S_3$ (sharding the second dimension of the second input, which corresponds to column tensor parallelism in Megatron-LM). The CombineSpec shows that, under $S_1$, we need to `gather` on the first dimension of the output; under $S_2$, we need to `reduce(SUM)` on the output; and under $S_3$, we need to `gather` on the second dimension of the output.

| MetaOp | MatMul: | | Elementwise-Add: | | LayerNorm: |
|---|---|---|---|---|---|
| **Input** | $X_1$ | $X_2$ | $X_1$ | $X_2$ | $X_1$ |
| | [ $D_1,D_2$ ] | [ $D_2,D_3$ ] | [ $D_1,D_2$ ] | [ $D_1,D_2$ ] | [ $D_1,D_2$ ] |
| **MetaSPMD** | *ShardSpec:* [ $S_1,S_2$ ], [ $S_2,S_3$ ] | | *ShardSpec:* [ $S_1,S_2$ ], [ $S_1,S_2$ ] | | *ShardSpec:* [ $S_1,N$ ] |
| | *CombineSpec:* [ $S_1$ : gather(dim=0), $S_2$ : reduce(op=SUM), $S_3$ : gather(dim=1) ] | | *CombineSpec:* [ $S_1$ : gather(dim=0), $S_2$ : gather(dim=1) ] | | *CombineSpec:* [ $S_1$ : gather(dim=0) ] |

Figure 3: MetaSPMD of some common operators.

MetaSPMD encompasses all the parallel strategies at the operator level, which can be utilized by automatic parallelism algorithms. Traditionally, automatic parallelism algorithms have required the manual annotating of this information for each operator within a specific IR. However, MetaDist utilize the ShardCombine Algorithm to automatically annotate MetaSPMD.

**Design of ShardCombine Algorithm.** The ShardCombine algorithm is an exploration algorithm that utilizes heuristic information to shard input data, and attempts to re-combine local results into global results using the `TryCombine` function. The high-level pseudo-code for this algorithm can be found in Algorithm 1. In the function `TryCombine(dims)`, the first step is to run `MetaOp.op_function` with `MetaOp.invars` as input to obtain global results. Next, the corresponding dimensions of `MetaOp.invars` are sharded, and the `op_function` is attempted to be executed to obtain local results. Finally, an attempt is made to obtain global results from local results using a predefined `CombineFunc`. If successful, the function returns the `CombineFunc`; otherwise, it returns `None`. It is important to note that many attempts to execute the `op_function` will fail due to shape mismatches, and these will immediately return a failure without the need to attempt combining.

---

**Algorithm 1:** ShardCombine Algorithm

**Input:** `MetaOp`
**Output:** `MetaSPMD` for this `MetaOp`

1 `MetaSPMD.ShardSpec ← [[NoShardDim] * var.ndim for var in MetaOp.invars]`
2 `MetaSPMD.CombineSpec ← {}`
3 `ShardID = 1`
4 **for** `dim` in `flatten(MetaSPMD.ShardSpec)` **do**
5     **if** `dim == NoShardDim` **then**
6         `try_assign ShardDim(ShardID)` on `dim`
7         **for** `dims` in the set of all subsets of `NoShardDim` in subsequent invars **do**
8             `combine_func = TryCombine(dim+dims)`
9             **if** `combine_func is not None` **then**
10                 `assign ShardDim(ShardID)` on `dim+dims`
11                 `MetaSPMD.CombineSpec[ShardID] = combine_func`
12                 `ShardID += 1`, **break**
13             **end if**
14         **end for**
15     **end if**
16 **end for**

Figure 4 illustrates an example of the ShardCombine algorithm applied to the matrix multiplication (MatMul) operator. Initially, we assign a value of `NoShardDim` ($N$) to each dimension of the input. We then attempt to assign a value of `ShardDim(1)` ($S_1$) to the first input dimension. Next, we check whether performing a `gather` operation on the first dimension of the output would result in a successful combination. On the second attempt, we determine that assigning $S_2$ to both the second dimension of the first input and the first dimension of the second input would enable us to execute the `op_function` and obtain global results by combining `reduce(SUM)`. The final step can be accomplished in a similar manner.

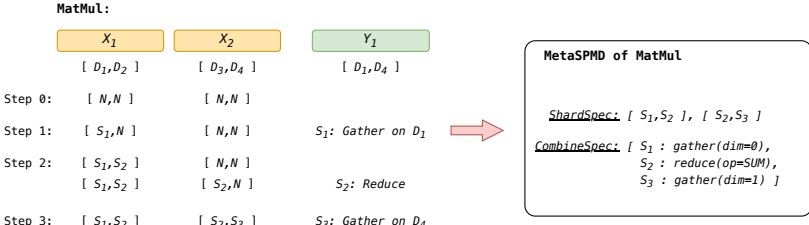

Figure 4: An example of the ShardCombine Algorithm on the matrix multiplication (MatMul) operator. The left side shows the exploration process of ShardCombine Algorithm, and the right side shows the MetaSPMD output of ShardCombine Algorithm.

**Implementation and Optimization of ShardCombine Algorithm.**

The *effectiveness* of this approach is the most critical question, as it determines whether it can encompass all operator-level parallelism strategies. As discussed in Section 3.1, the majority of operator-level parallelism can be expressed through shard, local computation, and re-combine. The ShardCombine algorithm effectively navigates this range of expression capabilities for each operator, enabling it to produce MetaSPMD for most operators in an efficient manner. Therefore, the ShardCombine algorithm's capability to include most operator-level parallelism strategies makes it a valuable approach.

We offer two methods to expand the functionality of the ShardCombine algorithm. Firstly, we can generate MetaSPMD directly for certain operators without undergoing the exploration process of the ShardCombine algorithm. This approach is useful for operators like reshape, which cannot be expressed as a simple shard and re-combine process due to the involvement of shape and stride derivation. Secondly, we can extend the algorithm's strategy space by expanding its shard and combine capabilities. For instance, we added support for the halo argument when dealing with convolution operators. In Figure 5, the gather function is extended with a halo argument. The halo argument can be a positive or negative number. If it is positive, we add the data in the overlap region when we combine the results. If it is negative, we discard the data with width $d$ and then perform the gather operation. By using such extensions, we can search for operator-level parallel strategies for convolution operators that involve sharding with the image length and width. Refer to Appendix A.3 for a demonstration of how to support complex cases in details

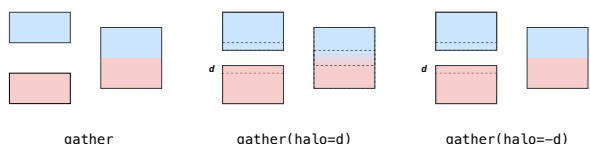

Figure 5: An example of extending `CombineFunc`. The left side shows the local results, the right side is the global results, different colors represent the data from different shards, the mixed color represents the sum of the data in the corresponding area.

Regarding the *efficiency* of the ShardCombine Algorithm, there are two aspects to consider. First, the time required for the ShardCombine Algorithm to explore all the operators is negligible compared to the training time of tens of hours or even days. This exploration process only needs to be done once before training starts. Second, our implementation uses a cache mechanism that leverages the `op_function` and `invars` as unique keys to find completed search strategies in the cache.

This mechanism can also speed up the process for a specific operator with different inputs. Since the MetaSPMD is approximately the same for an operator, even with different inputs, historical exploration results can be used as the initial state to reduce the number of attempts. Therefore, the ShardCombine Algorithm can significantly improve the efficiency of training deep neural networks.

The cache mechanism can greatly reduce search time, but its effectiveness varies depending on the model's characteristics. In the case of transformer architecture models, the cache mechanism is particularly effective for reducing search costs in the ShardCombine algorithm due to the model's small number of operators and fixed tensor shape.

### 3.4 BASELINE AUTOMATIC PARALLELISM ALGORITHM

We have implemented two baseline automatic parallelism algorithms using the MetaDist infrastructure to validate its effectiveness. One algorithm is based on integer linear programming (ILP), while the other is designed based on beam search. With the support of MetaDist, we have created a platform-agnostic implementation of automatic parallelism algorithms that can natively support PyTorch and Jax, without the need for manual annotation for operator-level SPMD rules. We have also used PyTorch's `DTensor` and Jax's `jax.Array` with `jax.lax.with_sharding_constraint` to inject the strategy generated by the automatic algorithm back into the framework via transformations. Finally, we have executed the computational graph using the framework's native runtime.

#### 3.4.1 AUTOMATIC PARALLELISM WITH ILP

In the context of MetaIR, we can view MetaOp as a node ($N$) and MetaVar as an edge ($E$), which collectively form a graph $G = (N, E)$. Following the approach proposed by Alpa Zheng et al. (2022) for intra-operator parallelism, we can model automatic parallelism as integer linear programming (ILP) problems and use an off-the-shelf solver to determine the optimal solution.

For a given node `node`$_i$, let `N`$_i$ denote the number of possible sharding strategies available. We define a decision vector $m_i \in \{0, 1\}^{N_i}$ for each node, where each element that takes a value of 1 in $m_i$ represents the selection of the corresponding sharding strategy. For edges connecting nodes `node`$_i$ and `node`$_j$, we use $M_{i,j} \in \{0, 1\}^{N_i \times N_j}$ as the decision vector. We also define a communication cost matrix $C_{i,j} \in \mathbb{R}^{N_i \times N_j}$. The objective of the problem is to minimize the communication cost $\sum_E \sum_{i,j} M_{i,j} * C_{i,j}$. The constraints for this problem are as follows:

- `sum(m`$_i$`) == 1` for each node (`m`$_i$ is one-hot vector)
- `sum(M`$_{i,j}$`) == 1` for ecah edge (`M`$_{i,j}$ is one-hot vector)
- for each edge connect `node`$_i$ and `node`$_j$
    - `M`$_{i,j}$ `<= m`$_i$`, M`$_{i,j}$ `<= m`$_j$
    - `M`$_{i,j}$ `>= m`$_i$ `+ m`$_j$ `- 1`

In contrast to Alpa, which is implemented at the XLA compiler level, our automatic parallelism algorithm is implemented on top of the ML framework. Therefore, we utilize Python-MIP Santos & Toffolo (2020) as the solver. For each node, we can generate all possible sharding strategies based on the MetaOp's MetaSPMD information and the current device mesh.

#### 3.4.2 AUTOMATIC PARALLELISM WITH BEAM SEARCH

In addition to ILP, we have also implemented another baseline automatic parallelism algorithm based on beam search. To implement the beam search algorithm, we loop through the `op_list` in MetaIR and maintain a strategy candidate set of size $K$. For each newly added $op_i$, assuming it has $n_i$ strategies, we combine each strategy in the current strategy candidate set with the strategy of this operator to generate a new set of strategies. If there are $N$ strategies in the current strategy candidate set, and after adding the new operator, we get $N * n_i$ strategies. From this new set, we filter out the optimal $K$ strategies based on the cost function. The beam search algorithm, like the ILP algorithm, aims to minimize communication as an optimization objective. However, the beam search algorithm relies on local information for decision-making, which can lead to getting stuck in local optima and makes it more challenging to consider memory constraints.

---

**Algorithm 2:** Automatic Parallelism with Beam Search

---

**Input:** `MetaIR`, `BeamWidth (K)`
**Output:** `ShardingStrategy` for this `MetaIR`

---

1  `CandidateSet ← []`
2  **for** `op` in `MetaIR.op_list` **do**
3    |  `CandidateSet = MERGE(CandidateSet, op.get_strategies())`
4    |  *// select the K strategies with less cost*
5    |  `CandidateSet = TopK(CandidateSet, K)`
6  **end for**
7  `ShardingStrategy = sorted(CandidateSet)[0]`

---

## 4  EXPERIMENTS

We conduct experiments to discuss and evaluate the following three points: 1) the time consumption of MetaSPMD annotation based on the ShardCombine algorithm; 2) the time consumption of two auto-parallelism baseline algorithms with different models and frameworks; 3) the performance comparison between the SOTA system and the baseline algorithm implemented with MetaDist.

The experimental platform comprises a GPU server equipped with eight 32GB V100 GPUs are equipped, interconnected via NVLink (hybrid cube-mesh topology). And in MetaDist, we use Py-Torch 2.0.1 and Jax 0.4.6. For benchmark, we use FairScale 0.4.13 and Alpa 0.2.3.

**The time consumption of MetaSPMD annotation.** As shown in Figure 6, we performed MetaSPMD annotation using the ShardCombine Algorithm for all operators in the GPT and ResNet He et al. (2016) models of four sizes under PyTorch and Jax. The lines represents the number of operators, and it can be seen that under the Jax framework, because of its smaller operator granularity, the number of operators required is much larger than that of PyTorch, which is more obvious on ResNet. Furthermore, the annotation process under PyTorch is much faster, because Jax has more operators and involves compilation overhead. With the cache mechanism, PyTorch can complete annotation in seconds and Jax can also complete annotation in minutes.

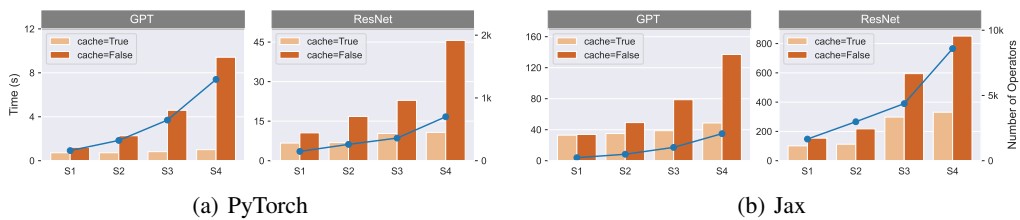

(a) PyTorch            (b) Jax

Figure 6: The bar chart represents the time consumption for MetaSPMD annotation of different models under PyTorch and Jax. $S1$ to $S4$ represent progressively larger models, and the blue lines and the right-hand Y-axis indicate their corresponding number of operators.

**The time consumption of two baseline auto-parallelism algorithms.** As shown in Figure 7, we performed two baseline algorithms on the GPT and ResNet models of four sizes under PyTorch and Jax. It can be seen that as the number of operators increases, the time for the auto-parallelism algorithm search all increases significantly. The ILP algorithm is able to complete the search within one or two minutes for models of different sizes. The BeamSearch algorithm is very fast when the number of operators is small. However, when the number of operators exceeds thousands, the overhead of its dictionary operations can cause a sharp increase in time consumption.

**Performance comparison with the SOTA system.** We chose three models for evaluation of the weak scaling, GPT Brown et al. (2020), WideResNet Zagoruyko & Komodakis (2016) and GAT Veličković et al. (2017) (see supplementary materials for configuration). GPT, WideResNet is a common evaluation workload for auto-parallelism algorithms. GAT is a graph neural network that extends the range of evaluation models in the future. Because the input of GAT is a graph containing its node features and adjacency matrix, it has no data dimension and cannot use even the most basic

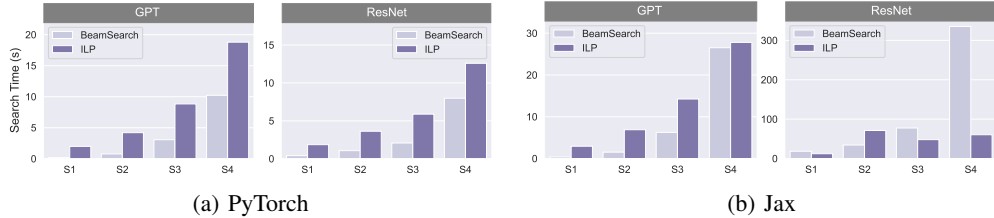

(a) PyTorch  (b) Jax

Figure 7: Time consumption of two automatic parallelism baseline algorithms for different models and sizes. The model size and the number of operators correspond to those in Figure 6.

data parallelism. For PyTorch, we use the PyTorch FSDP implementation of ZeRO-3 Rajbhandari et al. (2020) as the baseline, and for GPT, we add the tensor parallelism implemented with FairScale FairScale authors (2021) as the baseline, and for Jax, we use Alpa Zheng et al. (2022) as the primary baseline (only intra-op parallelism here). We use the aggregated floating-point operations per second (Flop/s) as the performance metric. It is worth noting that the ML framework uses the winograd convolution algorithm Lavin & Gray (2016), which requires fewer Flop than the theoretical estimate, so it can lead to higher metrics in WideResNet.

As shown in Figure 8, data parallelism cannot be adapted to model enlargement and soon OOM. ZeRO-3 is difficult to scale efficiently because of its communication overhead. On GPT, the ILP implemented based on MetaDist can basically reach the performance of hand-optimized (Tensor-Parallel). And in the Jax, it can be roughly reach the performance of the auto-parallel system Alpa, with some slight performance advantages on 2 and 4 GPUs. Furthermore, MetaDist-ILP performs best in all cases of GAT. On the other hand, beam search (MetaDist-BS) is less effective than the ILP algorithm because it tends to fall into local optima, and is more likely to exceed memory limits.

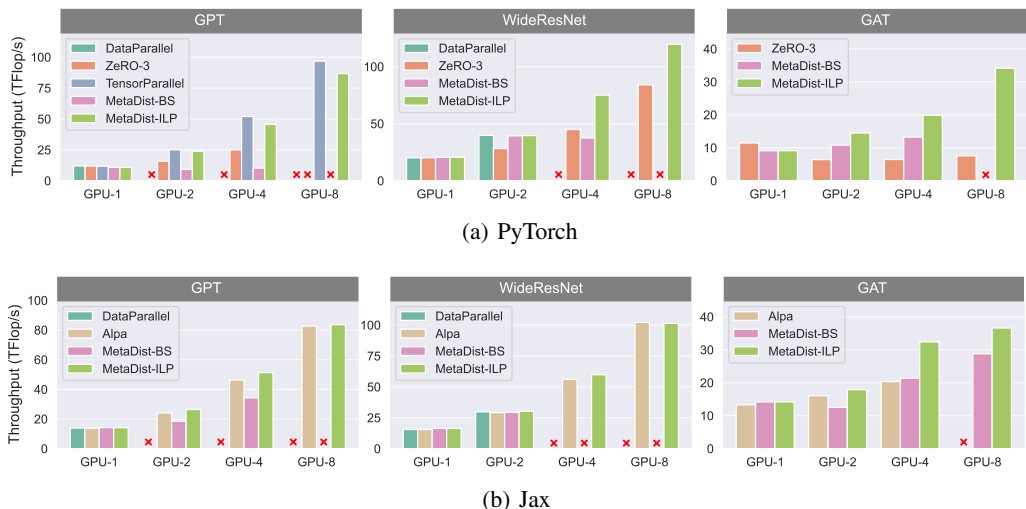

(a) PyTorch

(b) Jax

Figure 8: Performance comparison of two auto-parallelism baseline algorithms implemented on MetaDist and other SOTA systems. `MetaDist-BS` and `MetaDist-ILP` represent the two auto-parallelism algorithms implemented with MetaDist. Red crosses represent out-of-memory (OOM).

## 5 CONCLUSION

In conclusion, the use of MetaOp, MetaIR, and the ShardCombine Algorithm for automatic parallelism provides a framework-agnostic approach for decoupling parallelism strategy and ML framework. These developments in ecological compatibility have the potential to significantly improve the efficiency and effectiveness of developing, maintaining, and benchmarking automatic parallelism. Our baseline implementation and experiments demonstrate that our approach achieves state-of-the-art performance compared with other distributed solutions.

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

## A APPENDIX

### A.1 FRAMEWORK INTEGRATE: TVM TENSOR EXPRESSION

To support TVM, we first need to specify that `tvm.nd.NDArray` is the data structure for Tensors in `metadist.platform.tvm`. Then, we must define and register basic operations such as `add`, `concatenate`, `chunk`, `allclose`, and others that are required to perform relevant calculations in the ShardCombine algorithm.

Here's an example of how we can use MetaDist to run the ShardCombine Algorithm on TVM:

```python
import tvm, metadist

n = tvm.te.var("n")
A = tvm.te.placeholder((n, ), name="A")
B = tvm.te.placeholder((n, ), name="B")
C = tvm.te.compute(A.shape, lambda i: A[i] + B[i], name="C")

s = tvm.te.create_schedule(C.op)
tgt = tvm.target.Target(target="llvm", host="llvm")
fadd = tvm.build(s, [A, B, C], tgt, name="myadd")

def fadd_wrapped(a, b):
    c = metadist.platform.tvm.zeros_like(a)
    fadd(a, b, c)
    return c

meta_op_ = metadist.unifyshard.MetaOp(fadd_wrapped, ((a, b), {}))
meta_op_.sharding_annotation()
```

Note that in MetaDist, we need to shard and combine the input and output. However, TVM's kernel includes the output in the parameters of the input. Thus, we need to wrap the TVM kernel with the `fadd_wrapped` function before using it. Once the kernel is wrapped, we can apply the MetaSPMD annotation to the function via `sharding_annotation`. Adding distributed support for TVM is mainly missing communication features and distributed runtime. we will try to fully support TVM in future work.

### A.2 DETAILED CONFIGURATION OF THE MODEL IN SECTION 4

For our weak-scaling experiments in Figure 8, we used three different models: GPT, WideResNet, and GAT. Regarding models for benchmarking, we carefully selected well-known models such as GPT and WResNet. These choices allowed us to conduct a scientific comparison with existing approaches effectively. Moreover, we intentionally included models like GAT, which lack designed parallelism and have not been extensively explored in the context of auto-parallelism.

The detailed configurations for each model are shown in Table 2, 3 and 4.

Table 2: Four sizes of GPT models for evaluation.

| Number of GPUs | Number of parameters (billion) | Number of layers | Hidden size | Attention heads | Batch Size | TFLOPs |
|---|---|---|---|---|---|---|
| 1 | 1.26 | 1 | 10240 | 40 | 8 | 27.75 |
| 2 | 2.52 | 2 | 10240 | 40 | 8 | 58.93 |
| 4 | 5.03 | 4 | 10240 | 40 | 8 | 121.29 |
| 8 | 10.07 | 4 | 14336 | 56 | 8 | 237.15 |

Table 3: Four sizes of WideResNet models for evaluation.

| Number of GPUs | Number of parameters (billion) | Number of layers | Width | Batch Size | #FLOPs (tera) |
|---|---|---|---|---|---|
| 1 | 0.63 | 50 | 448 | 64 | 37.08 |
| 2 | 0.63 | 50 | 448 | 128 | 74.17 |
| 4 | 1.23 | 50 | 320 | 128 | 145.22 |
| 8 | 1.23 | 50 | 320 | 256 | 290.44 |

Table 4: Four sizes of GAT models for evaluation.

| Number of GPUs | Number of parameters (billion) | Number of nodes | Hidden size | Number of heads | #FLOPs (tera) |
|---|---|---|---|---|---|
| 1 | 0.6 | 1024 | 24576 | 1 | 2.63 |
| 2 | 1.21 | 1024 | 24576 | 2 | 6.49 |
| 4 | 2.42 | 1024 | 24576 | 4 | 14.23 |
| 8 | 4.83 | 1024 | 24576 | 8 | 29.69 |

### A.3 DETAILS ABOUT GATHER

What's challenging and interesting is the complexity of the different operation. MetaDist introduces two arguments in Gather, halo and block to support operators, such as convolution and concat. These two operators are the more frequently used operators in deep learning models. We found that the normal ShardCombine approach cannot describe these two operators. So we supported them by extending the Gather function. And because the complete exploration of the space is large and time consuming, we use some prior knowledge for efficiency.

#### A.3.1 HALOGATHER AND CONVOLUTION

In Figure 9, the shard and gather is extended with a halo argument. In the left case, we . If it is positive, we add the data in the overlap region when we combine the results. If it is negative, we discard the data with width $d$ and then perform the gather operation. In this case

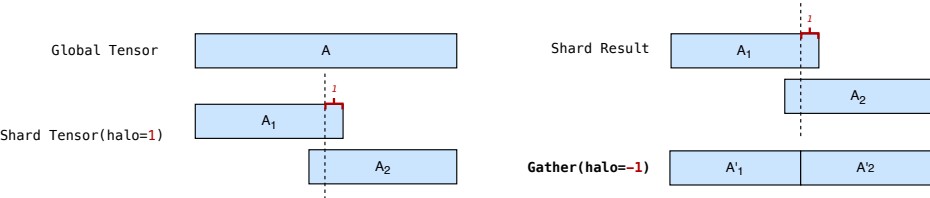

Figure 9: Meaning of the argument *halo* in Shard and Gather

Figure 10 illustrates the two most common convolution operators. The left panel displays $convolution(kernel = 3, pad = 0)$, while the right panel shows $convolution(kernel = 3, pad = 1)$. For simplicity and ease of understanding, we use 1D convolution. Similar methods can be used in 2D and 3D convolution. In both cases, we assume that the input Tensor is $A$, which has $x$ elements. After shard, each of our two devices contains half of the elements of $A$. Then we perform a local convolution calculation. In convolution without padding, the halo argument can be inferred from the tensor size. In convolution with padding, since the last row of data is computed under ZERO padding and is not equivalent with the original computation, the size of the shard halo can be inferred from the `allclose_rows`. And when trying to GATHER, halo argument of GATHER can be inferred from the tensor size.

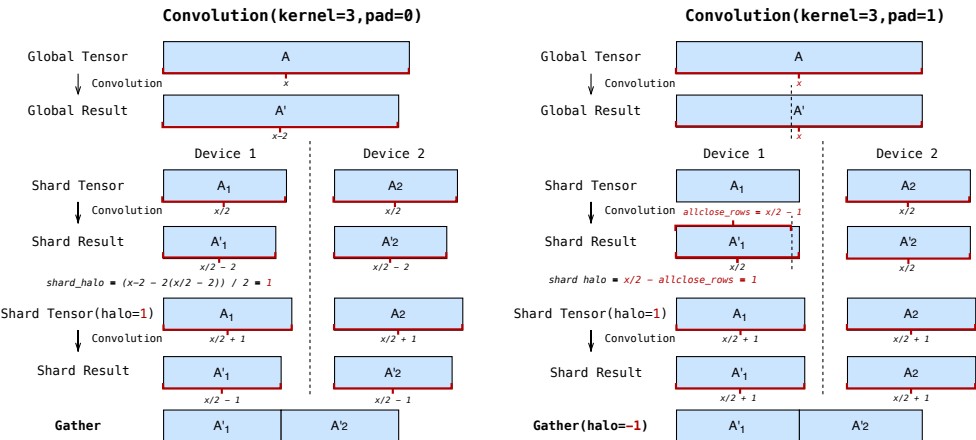

Figure 10: Two kinds of convolution (commonly used) are shown.

### A.3.2 BLOCKGATHER AND CONCAT

The concat (or concatenate) operator accepts a set of tensors as input, and its output is a tensor created by splicing together the input tensors. We can observe that if we shard each input tensor and then concatenate them locally, the results do not align with the global result; it resembles a block-cyclic distribution. Therefore, we introduce the argument named `block`. The gather operation, with $block = n$, first divides the shard tensor into $n$ parts, then performs `all-gather` on each part, and finally splices the results of the $n$ `all-gather` parts.

Figure 11 shows depicts a concatenation of three tensors, A1, A2, B1, B2, C1, C2 represent the shards of these three tensors. After concatenating them locally, we observed that only the first `allclose_rows` elements could be aligned. Therefore, we can infer from this information that 'block' is set to 3, meaning that the combine function here is `Gather(block=3)`.'

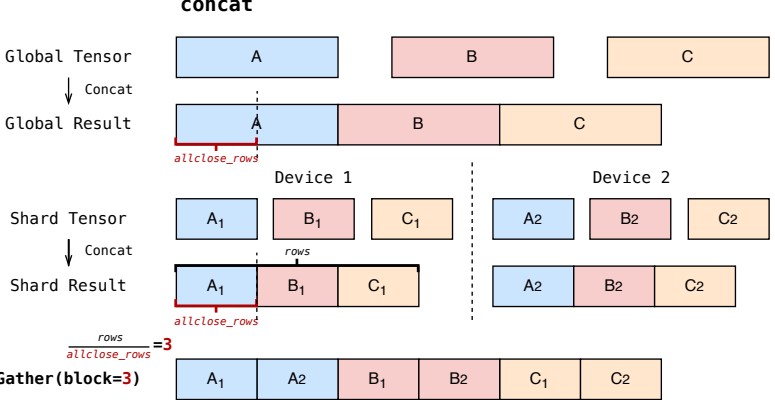

Figure 11: This example shows the concat operator concatenating three tensors. `rows` represents the number of rows in the first shard result. `allclose_rows` represents the number of rows in the first shard result that are allclose with the global result. Dividing the two yields the guessed argument of block, which is used to try and validate.

