# OpenReview forum: "MetaDist: An Infrastructure for Automatic Parallelism via ShardCombine Algorithm"
_ICLR.cc/2024/Conference — Submitted to ICLR 2024_

### Official Review · Reviewer_brSL · 2023-10-25

**Soundness:** 2 fair
**Presentation:** 3 good
**Contribution:** 2 fair
**Rating:** 3
**Confidence:** 3

**Summary:**

As the size of models becomes larger, distributed training has been paid much attention to. In this paper, the authors develop an infrastructure for automatic parallelism. Their infrastructure is based on two data structures and compatible with multiple ecologies, including PyTorch and JAX. The performance of their solution is better than existing methods.

**Strengths:**

1. The problem investigated in this paper is fundamental.
2. The presentation of this paper is clear. It is easy for me to follow the paper.
3. The infrastructure has great performance.

**Weaknesses:**

1. The main issue of this paper is the motivation of this paper is not clear. Since we already have the Alpa system to do distributed training automatically, I do not know why we need to develop a new tool. In addition, the experimental results show that the infrastructure developed in this paper has similar results as Alpa over GPT and WideResnet. Furthermore, the GAT model is not large enough. In most cases, we use the GAT model to solve small-size graphs. Therefore, I believe that Alpa is good enough.
2. As for automatically distributed training, the search time is an important metric. Is it possible to compare the search time of MetaDist with Alpa in the experiment section?
3. Could the authors evaluate their infrastructure over multiple nodes?
4. Megatron-LM is a widely used distributed training framework. I think the authors also need to add Megatron-LM as baseline for Pytorch implementation.
5. As an infrastructure, I think the authors should open their source code. If this infrastructure is used by many researchers and engineers, it means that this infrastructure is useful. In view of this, the GitHub star and download times are two important metrics for an open-sourced infrastructure.
6. This paper looks like a system paper, and I am not sure whether this paper could be accepted by the ICLR community.

**Questions:**

Please see the Strengths and Weaknesses.

---

### Official Review · Reviewer_9K4K · 2023-10-29

**Soundness:** 3 good
**Presentation:** 3 good
**Contribution:** 2 fair
**Rating:** 3
**Confidence:** 4

**Summary:**

This work proposes MetaDist, which provides automatic parallelism for different frameworks such as PyTorch and Jax.

**Strengths:**

I appreciate the engineering effort made in this work and it is really thrilling to see an automatic parallelism toolkit that is framework agnostic. I believe this is significant for the industry to ease the difficulty of training large models.

**Weaknesses:**

(W1) Limited contributions. It is unclear how the proposed MetaDist representation and the ShardCombine algorithm improve over existing works.

(W2) This work does not support pipeline parallelism. And the searching algorithm does not take computation cost into account.

(W3) More baselines are expected and some of the experimental results need elaboration.

**Questions:**

(Q1) From my humble opinion, this work is not well motivated and the rationality behind the proposed methods is not well explained.
- First, in the introduction section, the authors discussed two main challenges, i.e., (i) lack of ecological compatibility, and (ii) difficulty in development, maintenance, and benchmarking. However, they seems more of an implementation issue rather than a research topic.
- Second, there is no discussion about the limitations of existing methods and how the MetaDist representation and the ShardCombine algorithm tackles them (see questions below). For me, they are just put forward straightforwardly.

(Q2) In essence, MetaDist can be viewed as a kind of IR in a distributed manner. Therefore, I believe it is necessary to compare MetaDist related works such as pONNX [1] and Unity [2]. For me, the ShardSpec and CombineSpec can be broken down into the partition, combine, replicate, and reduce operators in Unity, and the latter can provide more fine-grained representations.

[1] Wang et al. Parallel Training via Computation Graph Transformation.\
[2] Unger et al. Unity: Accelerating DNN Training Through Joint Optimization of Algebraic Transformations and Parallelization.

(Q3) MetaDist focuses on intra-op parallelism, so it is unknown whether could it support pipeline parallelism, which is frequently used in large-scale distributed deep learning.

(Q4) Section 3.3 states that “The ShardCombine algorithm is an exploration algorithm
that utilizes heuristic information to shard input data, and attempts to re-combine local results
into global results using the TryCombine function”. What is the heuristic information here? Furthermore, how the heuristic information helps your algorithm design?

(Q5) Section 3.4.1 states that “The objective of the problem is to minimize the communication cost”. It is confusing why computation cost is not taken into account here. Furthermore, it is unclear how the ILP differs from the one in Alpa.

(Q6) The experiments were conducted on a single GPU server with eight GPUs. I am afraid the exploration space of parallelism would be extremely limited and could not evaluate the performance thoroughly.

(Q7) Regarding the experimental results, there are a few issues:
- The major competitors are Alpa and FairScale, however, there should be frameworks that are more widely used for large model training in practice, such as DeepSpeed and Megatron-LM. I believe comparisons are necessary.
- The authors stated that “ZeRO-3 is difficult to scale efficiently because of its communication overhead”. When memory is not scarce, ZeRO-2 may be a better option than ZeRO-3 and pure DataParallel. Please also compare with ZeRO-2.
- When training on a single GPU, there should be no performance differences, but they are reported in Figure 8. Please elaborate.
- There are several cases that MetaDist-BS runs out of memory. Does it indicate that your searching algorithm does not take memory limit as the constraint?

---

### Official Review · Reviewer_Rppy · 2023-11-01

**Soundness:** 2 fair
**Presentation:** 2 fair
**Contribution:** 2 fair
**Rating:** 3
**Confidence:** 5

**Summary:**

This manuscript / work is investigating the automatic parallelism and in particular is concerned with the selection of the best combination of strategies from a selection space of parallel strategies. The authors present the MetaDist, which they claim it is an infrastructure for automatic parallelism. They propose two abstract data structures, the MetaOp and MetaIR, respectively, which enable them to construct the MetaSPMD space.
The ShardCombine Algorithm obviates the need for manual annotation, significantly reducing the development and maintenance cost. Moreover, their approach is natively compatible with multiple ecologies, including PyTorch and JAX. To validate their design, they implement two baseline automatic parallelism algorithms based on MetaDist. Their experiments demonstrate that our approach achieves state-of-the-art performance compared with other distributed solutions.

**Strengths:**

+ Automatic parallelism is a crucial problem for current computing systems and especially recent machine learning architectures

**Weaknesses:**

- The impact of pruning on the prediction results needs to be addressed. How does pruning influence the accuracy or effectiveness of the model?
- When measuring throughput, it should be ensured that the profiling time and the time taken for pruning do not adversely affect the overall performance of the framework. The authors should explain how they have accounted for these factors in their evaluation.
- Incomplete discussion of prior work on automatic parallelization especially those that consider compiler approaches that are independent of a specific processor architecture.

**Questions:**

1) In section 3.2, what is callable primitive operator stans for in op_function, invars, outvars, and spmd_rules?
2)In figure 2, what is the MetaIR? There is a MetaIR on the left side and another MetaIR in the middle.
3) In Section 3.3 Sharedim(id=j)(Sj) is used to determine the parallelism strategies. How many such strategy could be applied? I notice the paper mentioned S1 S2 S3 in the rest of the paper. For Si, how many ways to partition the tensor?
4) What is the definition of score function? Is it just the train time? If two parallelisms could be applied, how to evaluate which one is better?
5) How is the communication bandwidth affect the result?
6) The treatment and discussion of prior work on automatic parallelism and compiler approaches for automatic parallelization only covers papers from 2022 and 2023 when there are pioneering approaches even before that. Here are some examples of compiler approaches to automatic parallelization: "A load balancing inspired optimization framework for exascale multicore systems: A complex networks approach." In 2017 IEEE/ACM International Conference on Computer-Aided Design (ICCAD), pp. 217-224. IEEE, 2017. "Self-optimizing and self-programming computing systems: A combined compiler, complex networks, and machine learning approach." IEEE transactions on very large scale integration (VLSI) systems 27, no. 6 (2019): 1416-1427. "A distributed graph-theoretic framework for automatic parallelization in multi-core systems." Proceedings of Machine Learning and Systems 3 (2021): 550-568. "Plasticity-on-chip design: Exploiting self-similarity for data communications." IEEE Transactions on Computers 70, no. 6 (2021): 950-962. The authors have to check the literature and carefully contrast all existing related approaches to automatic parallelization and compiler approaches especially when these prior approaches were developed to be computer architecture independent and adaptable to general settings.

---

### Official Review · Reviewer_KMij · 2023-11-01

**Soundness:** 3 good
**Presentation:** 2 fair
**Contribution:** 2 fair
**Rating:** 6
**Confidence:** 4

**Summary:**

The authors propose a technique to help address and improve the manual efforts necessary to parallelize training pipelines for large models on limited hardware. Two data structures MetaOps and MetaIR are proposed to construct the MetaSPMD space. MetaIR and MetaOp are framework-agnostic approaches where the computational graph of a network can be converted into MetaIR and the corresponding operators are converted to MetaOps. MetaSPMD specifies the operator-level parallel space of MetaOps and details how to shard inputs and combine local results into global ones.
Based on these structures, the paper proposes the ShardCombine algorithm to successfully shard input data and re-combine local results.

**Strengths:**

+ The ideas presented are well thought out, and much needed
+ Good that it works on both PyTorch and JAX
+ Experimental results could improve, but are not a bottleneck for an acceptance

**Weaknesses:**

- It might help to have some examples when the cache mechanism lacks effectiveness and is effective.
- The major issue with this is reproducibility. Simply by reading the paper, it will be difficult for me to reproduce the work in this paper. Artifacts will help. But the parts/sections about the MetaIR, MetaOps, and MetaSPMD could have been written so that one can understand how they are being practically generated. Conceptually, it’s fine.
- Why does Beam Search + ResNet + JAX in Figure 7 so much longer than any other combination? Is it simply because JAX has more fine-grained operators?
- The wiring could improve a bit, especially Sections 3 and 4.
It would have been nice to have some results on at least a 1 billion scale model. Or do the four different model sizes refer to GPT2, 3.5, etc.

**Questions:**

Please check the  Weaknesses for detailed questions to be answered.
- Are there some examples when the cache mechanism lacks effectiveness and is effective?
- Are there artifacts available for reproducibility purposes?
 - Why does Beam Search + ResNet + JAX in Figure 7 so much longer than any other combination? Is it simply because JAX has more fine-grained operators?
- Are there results on at least a 1 billion scale model? do the four different model sizes refer to GPT2, 3.5, etc?

---

### Meta-Review · Area_Chair_BQix · 2023-12-04

**Metareview:**

All reviewers had issues with this paper.  The main issues are clear motivation, clear description of novelty over prior work and clear demonstration that the results are reproducable.

**Justification For Why Not Higher Score:**

The reviewers have not identified key reasons why this paper should be accepted. Without interest, the paper is not ready for acceptance at a major conference.

**Justification For Why Not Lower Score:**

N/A

---

### Decision · Program_Chairs · 2024-01-16

Reject